# The effect of core stability training on ball-kicking velocity, sprint speed, and agility in adolescent male football players

Ceyda Sofuoğlu[1]☉*, Zehra Güçhan Topçu[2], Volga Bayrakcı Tunay[3]☉

1 Fizyo & Osteopathy Physical Therapy and Healthy Life Center, Nicosia, Cyprus, 2 Faculty of Physical Therapy and Rehabilitation, University of Hacettepe, Ankara, Turkey, 3 Faculty of Health Sciences, Physical Therapy and Rehabilitation Department, Eastern Mediterranean University, Famagusta, Cyprus

☉ These authors contributed equally to this work.
* ceyda_sofuoglu@hotmail.com

**Data Availability Statement:** All relevant data are within the manuscript and its Supporting Information files.

## Abstract

This study was conducted to investigate the effect of core stability training program on ball-kicking velocity, running speed, and agility in adolescent male football players. To this end, 36 male football players aged 12–14 were divided into the training group and control group. Before implementing the core stability training program, participants' ball-kicking velocity, sprint speed, and agility performance were measured with a Bushnell velocity radar gun, 20 m sprint test, and 505 agility test, respectively, in both training group and control group. After the measurements, the training group followed core stability training program three days a week, for eight weeks in addition to the routine training program, while the control group followed the routine training program only. Ball-kicking velocity, sprint speed, and agility performance were re-measured in both groups after the completion of the core stability training program. Significant improvements were detected in all parameters, i.e., ball-kicking velocity, sprint speed, and agility performance in the training group (p<0.05). On the other hand, in the control group, no significant change was detected in the ball-kicking velocity and sprint speed (p>0.05), whereas a significant improvement was observed in agility performance (p<0.05). Inter-group comparisons revealed statistically significant differences between the groups in ball-kicking velocity and sprint speed in favor of the training group (p<0.05), while no significant difference was found between the groups in agility performance (p>0.05). In conclusion, this study's findings suggest that core stability exercises can be incorporated into the routine training program of adolescent male football players.

## Introduction

Football is a game based on muscular performance, requiring the player to perform repetitive and high-intensity movements such as sudden acceleration and deceleration, changing direction, jumping, and landing [1]. It involves intermittent physical activities necessitating intense running in different directions during the game. Although football mainly involves running, explosive moves such as sprinting, jumping, and kicking the ball are required [1,2].

**Funding:** The author(s) received no specific funding for this work.

**Competing interests:** The authors have declared that no competing interests exist.

In adolescent football players, overuse injuries occur at a rate of 10–40%. The majority (60–90%) of injuries are observed in the lower extremity, including the ankle and thigh. The most common types of injuries are strains, sprains, and contusions [3,4]. Approximately 40%–60% of injuries occur due to contact with another player or object during the game [5]. Consequently, adolescents have an increased risk of injury from repetitive mechanical stress [6].

The core muscle group is defined as a muscle corset consisting of the abdominals in the front, the erector spinae and gluteals in the back, the diaphragm as the roof, and the pelvic floor and hip girdle muscles at the bottom [7]. These muscles collectively support the spine and trunk during upper and lower extremity movements such as jumping, throwing, running, and kicking the ball [8]. Kibler [9] explains core stability based on the principle of "proximal stability for distal mobility." According to this principle, optimal transfer and control of power and movement transmitted to the distal segments during sports activities are achievable through the ability to control the position and movement of the trunk on the lower extremities and pelvis [9]. Brown [7] adds that core stability is achieved through the dynamic restriction provided by the core muscles and the passive stiffness contributed by the vertebrae, fascia, and ligaments [11]. Panjabi [10] further notes that, in addition to bone, ligament, and muscle structures, the central nervous system also plays a role in contributing to core stability by providing neuromuscular control [10].

Training the core muscles is expected to enhance athletic performance and reduce injuries [9,11]. Studies indicate that disorders in core muscle strength and endurance, proprioception, and neuromuscular control of core muscles, which can affect core stability, are important risk factors in the development of lower extremity injuries [12,13]. The effect of core stability training (CST) on athletic performance has been and continues to be studied. Previous studies revealed that a CST program increased some parameters of general performance and sport-specific performance in badminton, handball, baseball, basketball, football, swimming, tennis, rhythmic gymnastics, and modern dance [14–23].

Although many studies in the literature investigate the effect of core exercise training on performance, only a few studies look into the effects of a CST program in adolescent football players [3,12]. We hypothesized that core stability exercises would improve ball-kicking velocity, sprint speed, and agility in adolescent male football players.

## Methods

### Experimental approach to the problem

The minimum number of participants required for a study power of 0.95 and alpha error 0.05 was determined using G Power 3.0.10 G Software for Windows (Heinrich-Heine-Universität Düsseldorf, Düsseldorf, Germany). The results revealed that 36 players are needed. To evaluate the effect of CST, 45 adolescent male football players who were on a similar training program, met the study criteria, and whose legal guardians signed the informed consent form were divided into the training group (TG) (n = 23) and control group (CG) (n = 22) (Fig 1). The TG followed the CST program three days a week for 8 weeks in addition to the routine training program, while the CG followed the routine training program only. The participants' ball-kicking velocity, sprint speed, and agility performances were evaluated by the same physiotherapist before training and eight weeks after training. Measurements data were collected at the beginning of the study and after the completion of the 8 weeks CST program. October 1, 2021 pre-study measurements were completed. CST program was implemented between the dates October 2, 2021- November 27, 2021. November 29, 2021 post-study measurements were repeated.

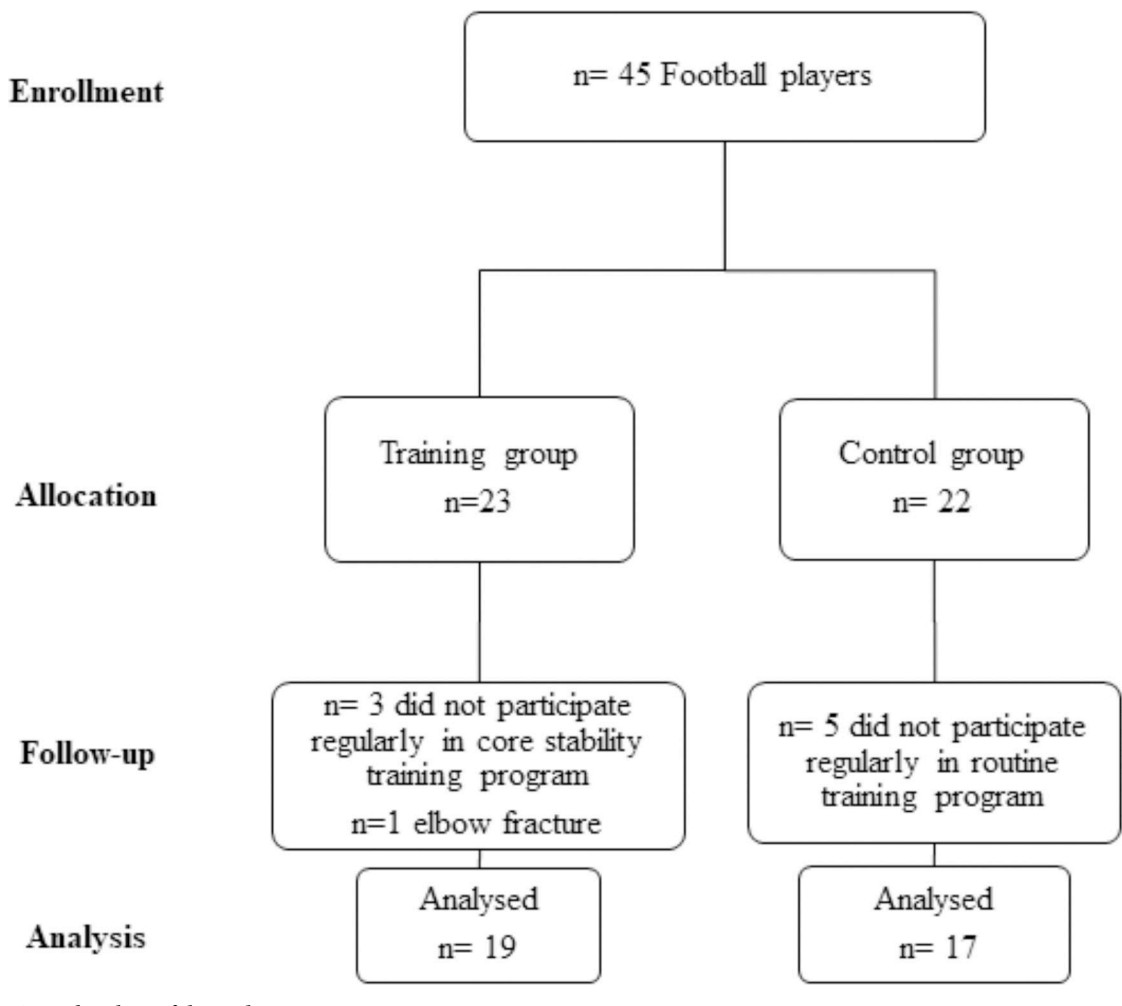

**Fig 1. Flowchart of the study.**

## Subjects

The protocol of this study was approved by the Eastern Mediterranean University Health Ethics Subcommittee Presidency prior to the conduct of the study. The study population consisted of male adolescent football players aged between 12–14, playing football at the youth setup of a football academy. The study inclusion criteria were determined as not having any pain complaints involving the lower extremities and spine, should had no surgery before of the study, not having a pathological condition, and participating regularly in training. On the other hand, the study exclusion criteria were determined as having taken a break from sports, not having participated in training more than three times, and having difficulty following exercise programs.

## Measurements

**Ball-kicking velocity test.** Kicking velocity was measured using a Bushnell velocity radar gun (Bushnell Performance Optics, Overland Park, KS, US) [24]. To this end, the ball was placed at a distance of 15 m from the goal. The tripod support of the velocity radar gun was set at a height of 1.22 m from the ground and kept 1 m behind the goal. The participants were

asked to run from a distance of 5 m and kick the ball with maximum force. A test kick was made, followed by three kicks. The average of these three kicks was recorded in km/h. In accordance with the International Federation of Association Football (FIFA) standards, 12-to-13-year-olds made the kicks with a number 4 soccer ball, and 14-year-olds made the kicks with a number 5 soccer ball [25,26]. Radar guns are a common tool both in practical use and scientific research due to their high accuracy (ICC> 0.94) [27,28].

**Sprint speed test.**   Sprint speed was measured via the 20 m sprint test. To this end, the participants were asked to start the sprint run 50 cm behind the starting line, and the time to complete 20 m was recorded in seconds. Sprint test was conducted using a stopwatch. The reliability of 20 m sprint test among U12 (ICC = 0.73), U13 (ICC = 0.90) and U14 (ICC = 0.83) has been demonstrated [29].

**Agility test.**   Agility performance was measured with the 505 agility test developed by Draper and Lancaster in 1985. To this end, the participants were first asked to complete a 10 m approach run, followed by moving back a distance of 5 m, making a 180˚ turn, and returning. The time it took for the participants to cover the said 5 m distance back and forth was recorded in seconds. Agility test was conducted using a stopwatch. The reliability of 505 agility test among U12 (ICC = 0.57), U13 (ICC = 0.91) and U14 (ICC = 0.89) has been demonstrated [29].

## Training program

Following the completion of performance assessment measurements, an exercise diary was created to track participants' attendance in the study. Subsequently, the TG followed the CST program in addition to the routine training program during the season for eight weeks, three days a week, under the supervision of the same physiotherapist. The program was implemented in three phases. Accordingly, phases 1, 2 and 3 were implemented between $1^{st}$–$3^{rd}$, $3^{rd}$–$5^{th}$, and $5^{th}$–$8^{th}$ weeks, respectively. Before the initiation of the program, transversus abdominis muscle and multifidus muscle contraction were explained and checked by palpation to enhance motor control, endurance, and kinesthetic awareness. This was followed by four sessions of individual training, including instructions on maintaining a neutral spine position, abdominal hallowing, and abdominal bracing techniques, utilizing a pressure biofeedback device. Ankle weights, dumbbells, resistance bands, and physio balls were incorporated into the program. The exercise equipments' used by participants were determined by verbally asking them about the weights they could lift ten times in a row without difficulty. Exercise intensity progressively increased every phases. From week 1–3 exercises were repeated 15 times, 1 set. From week 3–5 exercises were repeated 12 times, 2 sets. From week 5–8 exercises were repeated 10 times, 3 sets. The rest period between sets was one minute in all training phases. The exercise program content was changed every phases; thus, duration of the sessions was initially 40 minutes and increased to 60 minutes in the following weeks. Warm-up exercises were performed for 10 minutes before each session, and a 10-minute cool-down program was applied at the end of each session. The details of the CST program are outlined in Table 1.

## Statistical analysis

The descriptive statistics obtained from the collected data were tabulated as mean ± standard deviation or median with minimum and maximum values, depending on the normal distribution characteristics of continuous (numerical) variables. Categorical variables were presented as numbers and percentages. Normal distribution characteristics of numerical variables were assessed using the Shapiro-Wilk, Kolmogorov-Smirnov, and Anderson-Darling tests. In comparisons of two independent groups, the independent samples t-test or Mann-Whitney U test was used for normally and non-normally distributed numerical variables, respectively. For

**Table 1. Core stability training program.**

| Core Stability Exercises Phase 1 (1–3 Weeks) |
| --- |
| 1- Supine abdominal draw in with knee to chest X15, 1 set |
| 2- Supine abdominal draw in with heel side X15, 1 set |
| 3- Supine abdominal draw in with double knee to chest X15, 1 set |
| 4- Supine twist with abdominal drawing X15, 1 set |
| 5- Side lying clam exercise X15, 1 set |
| 6- Side lying leg-arm ipsilateral abduction X15, 1 set |
| 7- Side bridging on elbow hold 10 seconds X15, 1 set |
| 8- Prone position superman hold 10 seconds X15, 1 set |
| 9- Prone position alternate superman hold 5 seconds X15, 1 set |
| 10- Quadruped position abdominal drawing hold 5 seconds X15, 1 set |
| 11- Quadruped alternate arm and leg lift X15, 1 set |
| 12- Quadruped knee lifts hold 5 seconds X15, 1 set |
| 13- Prone plank on feet hold 5 seconds X15, 1 set |
| 14- Supine bridge with arms at side hold 5 seconds X15, 1 set |
| 15- Supine bridge with arms across chest hold 5 seconds X15, 1 set |
| 16- Supine bridge with knee extension hold 5 seconds X15, 1 set |
| 17- Supine bridge with single leg marching (hip and knee bent to 90 degrees) hold 5 seconds X15, 1 set |
| 18- Seated on physioball abdominal drawing hold 5 seconds X15, 1 set |
| 19- Seated on physioball add marching hold 5 seconds X15, 1 set |
| Core Stability Exercises Phase 2 (3–5 Weeks) |
| 1-Supine dead bugs X12, 2 sets |
| 2-Rolling like a ball hold 2 seconds in starting position X12, 2 sets |
| 3- Side bridge on feet with hip abduction X12, 2 sets |
| 4- Prone bridge on elbows with hip extension X12, 2 sets |
| 5- Prone bridge on hands, start with quadruped knee lift position X12, 2 sets |
| 6- Quadruped alternate arm/leg lifts with cuff and dumbbell weight X12, 2 sets |
| 7- Abdominal crunches on physioball hold 2 seconds X12, 2 sets |
| 8-Abdominal crunches on physioball with rotation hold 2 seconds X12, 2 sets |
| 9- Bridging with heaf on physioball hold 5 seconds X12, 2 sets |
| 10- Supine bridging on physioball hold 5 seconds X12, 2 sets |
| 11- Supine bridging on physioball with arms in 90 degrees' shoulder flexion and alternate leg lifts X12, 2 sets |
| 12- Straight leg lowering/lifting with physioball between legs X12, 2 sets |
| 13- Physioball decline push-up X12, 2 sets |
| 14- Seated on physioball with leg extension hold 5 seconds X12, 2 sets |
| Core Stability Exercises Phase 3 (5–8 Weeks) |
| 1- Supine bridge with feet on physioball hold 5 seconds, add alternate knee extension X10, 3 sets |
| 2- Physioball jackknife X10, 3 sets |
| 3- Prone position on physioball catching and throwing a football ball with partner X10, 3 sets |
| 4- Side position on physioball catching and throwing a football ball with partner X10, 3 sets |
| 5- Standing upright position hip flexion, extension, abduction and adduction with elastic resistance band X10, 3 sets |
| 6- Standing upright position catching and throwing a physioball with partner X10, 3 sets |
| 7- At squat position catching and throwing a football ball with partner X10, 3 sets |
| 8- At single leg stance catching and throwing a physioball with partner X10, 3 sets |
| 9- Walking lunges with catching and throwing a football ball X10, 3 sets |
| 10- Simultaning to kick with inside of foot while catching a physioball (elastic resistance band added) X10, 3 sets |
| 11- Simultaning to kick with outside of foot while catching a physioball (elastic resistance band added) X10, 3 sets |
| 12- Simultaning to kick with top of foot while catching a physioball (elastic resistance band added) X10, 3 sets |
| 13- Kicking a football ball inside, outside and top of the foot with partner (elastic resistance band added) X10, 3 sets |

comparisons of two dependent groups, the paired samples t-test was applied. Statistical analyses were conducted using Jamovi project 2.2.5.0 (Jamovi, version 2.2.5.0, 2022, retrieved from https://www.jamovi.org) and JASP 0.16.1 (Jeffreys' Amazing Statistics Program, version 0.16.1, 2022, retrieved from https://jasp-stats.org) software packages. A probability ($p$) value of $\leq 0.05$ was deemed to indicate statistical significance.

## Results

During the study period, four and five participants from the TG and CG groups, respectively, were excluded from the study. Three players from the TG did not participate regularly in CST and one player had an orthopedic injury. Five participants from CG did not participate regularly in routine training program. In the end, 19 and 17 participants completed the study in the TG and CG, respectively (Fig 1). The age, height, weight, body mass index, and sports age data of the participants showed a homogeneous distribution. The distribution of the participants' demographic characteristics by the TG and CG is given in Table 2.

The inter-group comparisons after the completion of the CST program revealed significant differences between the TG and CG. Accordingly, a significant difference was found between the TG and CG in ball-kicking velocity after the completion of the program, in favor of the TG ($p = 0.002$, $d > 1$). Similarly, the sprint speed test time was significantly higher in the CG than in the TG after the completion of the program ($p = 0.001$, $d > -1.2$). There was a significant difference between the TG and CG in the median percent change in sprint speed, in favor of the CG ($p = 0.006$). In parallel, the decrease in the time to complete the sprint speed test was significantly higher in the TG than in the CG (-5.56 [-21.03–4.13] sec. vs. -0.71 [-7.19–17.62] sec.). The inter-group comparisons in agility performance parameters did not reveal any statistically significant difference between the groups ($p > 0.05$). Effect size for time agility was $d = -0.51$ (CI = [-1.175–0.156]) (Table 3) (Figs 2–4).

The intra-group comparisons revealed significantly higher ball-kicking velocity and significantly lower sprint speed and agility test times with large effect sizes in the TG ($p = 0.002$, $d = -0.83$; $p < 0.001$, $d = 0.90$; $p < 0.001$, $d = 0.95$ respectively) (Table 4) (Figs 2–4).

**Table 2. Demographic characteristics data for training and control group.**

|  | Training group (n = 19) | Control group (n = 17) | Test statistic | Cohen's d [95% CI] | p-value |
|---|---|---|---|---|---|
| **Age** | 13.05 ± 0.78 | 12.82 ± 0.81 | 135.500 | 0.161 [-0.217–0.497] | 0.381** |
|  | 13.00 [12.00–14.00] | 13.00 [12.00–14.00] |  |  |  |
| **Sports age** | 6.11 ± 2.26 | 5.24 ± 2.39 | 1.124 | 0.375 [-0.288–1.033] | 0.269* |
|  | 6.00 [1.00–10.00] | 5.00 [2.00–9.00] |  |  |  |
| **Body weight (kg)** | 54.00 ± 9.71 | 52.59 ± 8.75 | 0.451 | 0.152 [-0.504–0.807] | 0.651* |
|  | 52.00 [38.00–78.00] | 51.00 [40.00–70.00] |  |  |  |
| **Height (cm)** | 161.21 ± 7.76 | 159.18 ± 9.28 | 0.716 | 0.336 [-0.420–0894] | 0.479* |
|  | 162.00 [144.00–176.00] | 158.00 [144.00–179.00] |  |  |  |
| **Body mass index (kg/m$^2$)** | 20.72 ± 3.13 | 20.72 ± 2.84 | 156.000 | 0.193 [-0.394–0.335] | 0.862** |
|  | 18.97 [17.30–28.31] | 21.08 [16.42–28.15] |  |  |  |

Descriptive statistics were expressed as mean ± standard deviation and median [minimum-maximum].

Test statistic: The numerical outcome of statistical tests comparing the demographic characteristics between the training and control groups.

CI: Confidence Interval.

p-values*: Results of Independent samples t-test.

p-values **: Results of Mann-Whitney u test.

**Table 3. Comparison of inter group performance tests before and after the core stability exercise program.**

| | Training group (n = 19) | Control group (n = 17) | Test statistic | Cohen's d [95% CI] | p-value |
|---|---|---|---|---|---|
| **Ball kicking velocity (km/hour)** | | | | | |
| Before the Program | 71.89 ± 16.21 | 67.71 ± 13.80 | 121.500 | 0.248 [-0.129–0.562] | 0.204** |
| | 78.00 [35.00–92.00] | 68.00 [29.00–88.00] | | | |
| After the Program | 81.05 ± 8.46 | 69.12 ± 13.04 | 3.292 | 1.099 [0.388–1.796] | **0.002*†** |
| | 82.00 [63.00–97.00] | 69.00 [32.00–89.00] | | | |
| Δ% | 7.14 [-8.75–87.50] | 4.41 [-14.71–19.61] | 100.000 | 0.381 [0.018–0655] | 0.051** |
| **Sprint speed (sec)** | | | | | |
| Before the Program | 3.86 ± 0.40 | 3.96 ± 0.25 | -0.897 | -0.300 [-0.956–0.361] | 0.376* |
| | 3.78 [3.28–4.81] | 4.00 [3.50–4.31] | | | |
| After the Program | 3.60 ± 0.23 | 3.95 ± 0.34 | -3.706 | **-1.237 [-1.947 –-0.513]** | **0.001*†** |
| | 3.59 [3.09–4.00] | 3.94 [3.44–4.94] | | | |
| Δ% | -5.56 [-21.03–4.13] | -0.71 [-7.19–17.62] | 74.000 | **-0.542 [-0.757 –-0.220]** | **0.006***** |
| **Agility (sec)** | | | | | |
| Before the Program | 3.03 ± 0.28 | 3.03 ± 0.20 | -0.017 | -0.006 [-0.660–0.649] | 0.987* |
| | 2.94 [2.66–3.60] | 2.96 [2.81–3.53] | | | |
| After the Program | 2.80 ± 0.28 | 2.93 ± 0.20 | -1.536 | -0.513 [-1.175–0.156] | 0.134* |
| | 2.75 [2.41–3.58] | 2.90 [2.68–3.40] | | | |
| Δ% | -4.82 [-21.94–4.07] | -3.68 [-9.37–10.33] | 122.000 | -0.245 [-0.560–0.133] | 0.211** |

Descriptive statistics were expressed as mean ± standard deviation and median [minimum-maximum].

Δ%: Percentage difference between two times (positive values indicate increase, negative values indicate decrease).

Test statistic: The numerical outcome of statistical tests comparing the inter group performance tests before and after the program between the training and control groups.

CI: Confidence Interval.

p-values

*: Results of Independent samples t-test.

p-values

**: Results of Mann-Whitney u test.

† Significate difference between training group and control group (p < 0.05).

On the other hand, in the CG, no significant change was detected in the ball-kicking velocity and sprint speed (p > 0.05, d = -0.24; p > 0.05, d = 0.02 respectively), whereas a significant improvement was observed in agility performance (p = 0.009, d = 0.71 CI = [0.171–1.242]) (Table 5) (Figs 2–4).

## Discussion

This study aimed to investigate the effects of core stability exercises on ball-kicking velocity, sprint speed, and agility in adolescent male football players. The findings of this study revealed that the implementation of a CST program, in addition to the routine training program during the season, is effective in improving ball-kicking velocity and sprint speed. On the other hand, the CST program did not lead to a significant improvement in agility performance among adolescent male football players. Although these results were generally consistent with previous studies conducted with athletes from different age groups and genders, as well as healthy individuals, they also exhibit some differences in terms of certain sport-specific parameters.

Kicking the ball, an action that engages multiple joints, stands out as one of the most crucial skills in football. The performance of ball-kicking relies on the maximum force and strength

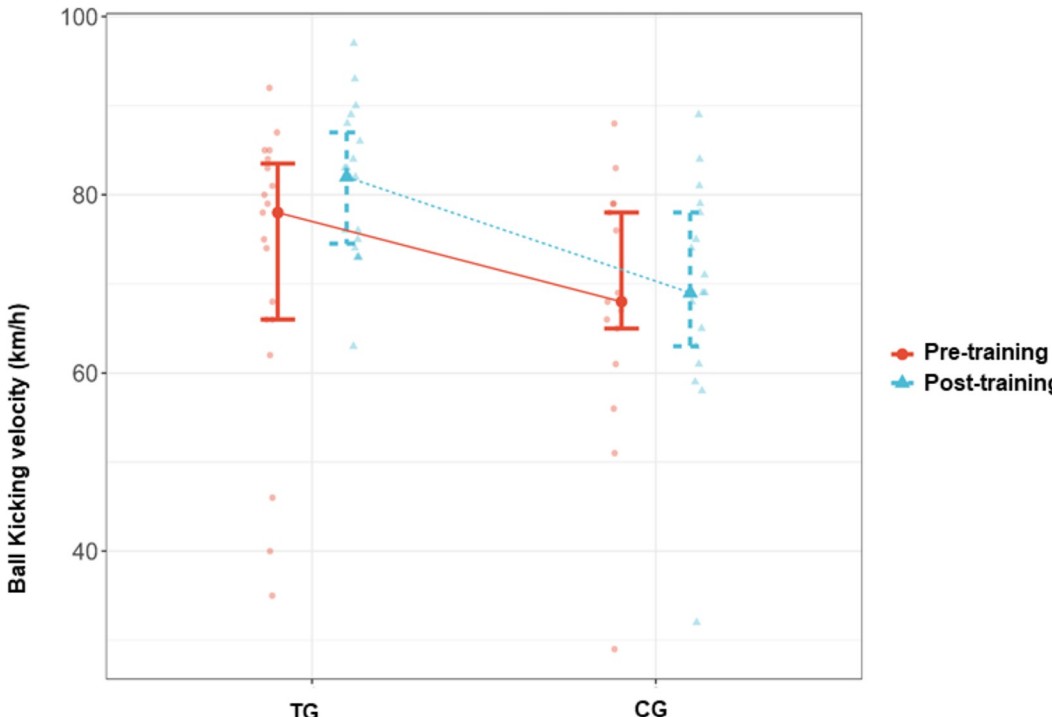

**Fig 2. Ball kicking velocity pre- training and post- training values in TG and CG.** TG = Training group, CG = Control group.

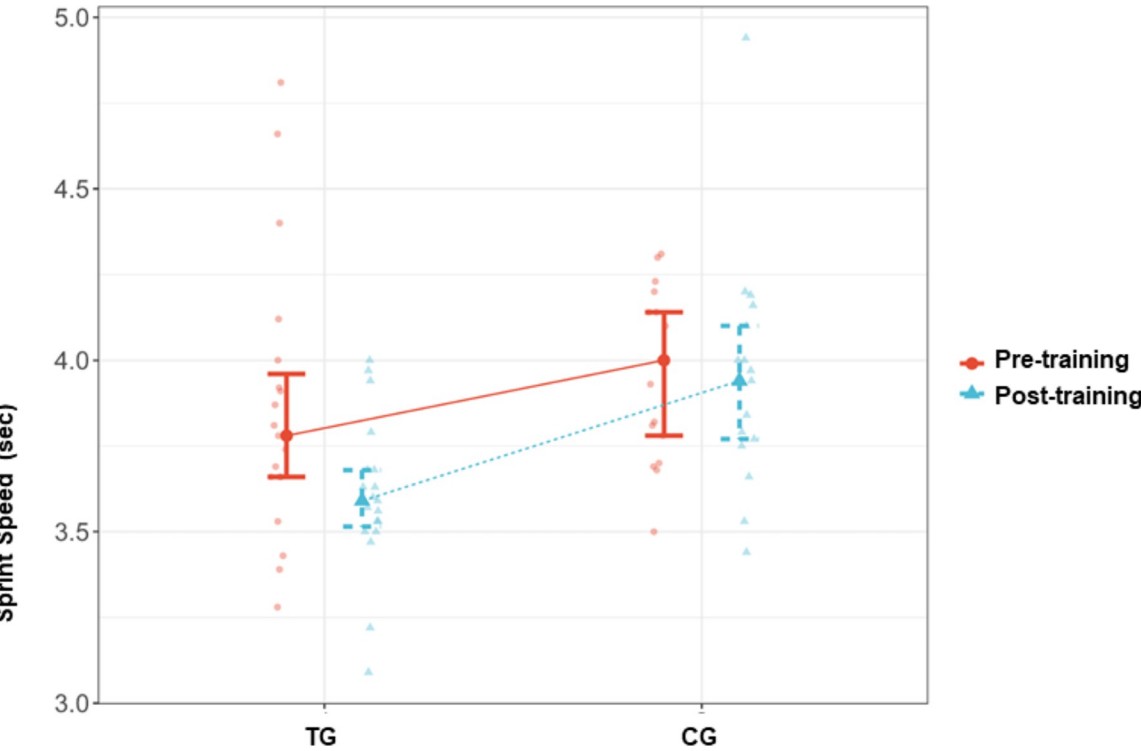

**Fig 3. Sprint speed pre- training and post- training values in TG and CG.** TG = Training group, CG = Control group.

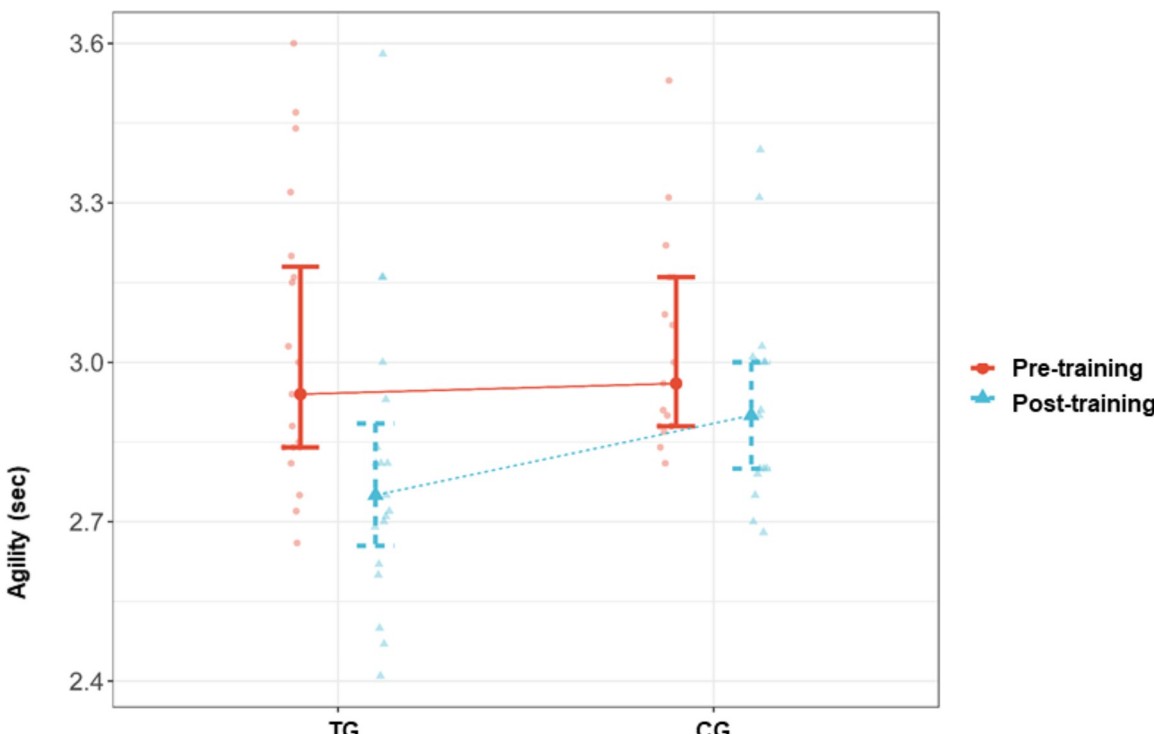

**Fig 4. Agility pre- training and post- training values in TG and CG.** TG = Training group, CG = Control group.

exerted by the foot responsible for both kicking the ball and providing support. Additionally, it depends on the coordination between agonist muscles (including vastus lateralis, medialis, rectus femoris, tibialis anterior, iliopsoas) and antagonist muscles (such as gluteus maximus, biceps femoris, and semitendinosus) [30,31]. Wickstorm explains kicking the ball in four phases: (1) the swing phase, where the thigh and shank are withdrawn; (2) the support phase, where hip flexion begins; (3) ball impact, involving knee extension; and (4) the follow-through, where both the hip and knee undergo flexion [25,32]. A review study, encompassing 96 studies on the relationship between ball-kicking velocity and muscle strength, unveiled a significant

**Table 4. Comparison of performance tests before and after the exercise program in the training group.**

| | Training group | | Test Statistic | Cohen's d [95% CI] | p-value |
|---|---|---|---|---|---|
| | Before the Program | After the Program | | | |
| **Ball kicking velocity (km/hour)** | 71.89 ± 16.21 | 81.05 ± 8.46 | -3.637 | -0.834 [-1.351 –-0.301] | **0.002**\*† |
| | 78.00 [35.00–92.00] | 82.00 [63.00–97.00] | | | |
| **Sprint speed (sec)** | 3.86 ± 0.40 | 3.60 ± 0.23 | 3.938 | 0.903 [0.358–1.431] | <**0.001**\*† |
| | 3.78 [3.28–4.81] | 3.59 [3.09–4.00] | | | |
| **Agility (sec)** | 3.03 ± 0.28 | 2.80 ± 0.28 | 4.148 | 0.952 [0.397–1.488] | <**0.001**\*† |
| | 2.94 [2.66–3.60] | 2.75 [2.41–3.58] | | | |

Descriptive statistics were expressed as mean ± standard deviation and median [minimum-maximum].

Test statistic: The numerical outcome of statistical tests comparing the performance tests before and after the exercise program in the training group.

CI: Confidence Interval.

p-values*: Results of paired samples t-test.

† Significant difference between before and after the program (p < 0.05).

**Table 5. Comparison of performance tests before and after the exercise program in the control group.**

| | Control group | | Test statistic | Cohen's d [95% CI] | p-value |
|---|---|---|---|---|---|
| | **Before the Program** | **After the Program** | | | |
| **Ball kicking velocity (km/hour)** | 67.71 ± 13.80 | 69.12 ± 13.04 | -0.995 | -0.241 [-0.720–0.245] | 0.335* |
| | 68.00 [29.00–88.00] | 69.00 [32.00–89.00] | | | |
| **Sprint speed (sec)** | 3.96 ± 0.25 | 3.95 ± 0.34 | 0.212 | 0.027 [-0.449–0.502] | 0.913* |
| | 4.00 [3.50–4.31] | 3.94 [3.44–4.94] | | | |
| **Agility (sec)** | 3.03 ± 0.20 | 2.93 ± 0.20 | 2.949 | 0.715 [0.171–1.242] | **0.009**\*† |
| | 2.96 [2.81–3.53] | 2.90 [2.68–3.40] | | | |

Descriptive statistics were expressed as mean ± standard deviation and median [minimum-maximum].

Test statistic: The numerical outcome of statistical tests comparing the performance tests before and after the exercise program in the control group.

CI: Confidence Interval.

p-values*: Results of paired samples t-test.

† Significant difference between before and after the program (p <0.05).

correlation between maximal isokinetic force and ball-kicking velocity in young football players. Conversely, the analysis of the results from explosive power tests and ball-kicking velocity in young football players did not reveal any significant relationship between maximal strength, explosive strength, and ball-kicking performance. They also found no significant relationship between the results of the 10 m sprint test and ball-kicking velocity. The aforementioned review study concluded that, despite the controversy surrounding the relationship between strength and ball-kicking velocity, incorporating plyometric and explosive strength exercises into the training routine may enhance maximum ball-kicking velocity [33]. Sporis et al. [26] investigated the relationship between ball-kicking velocity and sprint speed with 27 football players with an average age of 15±2.9. They measured ball-kicking speed using a velocity speed gun and running speed through 5, 10, 20, and 30 m sprint tests. In the mentioned study, in contrast to Lorenzo et al.'s [33] study, ball-kicking velocity was found to be significantly related to running speed [26]. In a study examining the relationship between ball-kicking speed and muscle strength, the dominant lower extremity strength of amateur football players was measured with an isokinetic dynamometer at 90˚ and 240˚ angular velocities. Consequently, it was demonstrated that participants with greater knee extension concentric strength had significantly higher ball-kicking speeds [34].

In our study, both intra- and inter-group analyses of ball-kicking velocity measurements, conducted with a velocity radar gun, revealed that the CST program significantly improved ball-kicking velocity. The noteworthy improvement in ball-kicking velocity within the TG compared to the CG may be attributed to the increase in muscle strength and postural control ability fostered by the CST program. Similarly, in a randomized controlled study examining the effect of in-season integrative neuromuscular strength training on performance development in early adolescent football players who followed a neuromuscular training program three times a week for eight weeks in addition to the routine training program, in comparison with a CG who only followed the routine training program, Panagoulis et al. [23] found significant improvements in all parameters. These included ball-kicking velocity, 10–20 m sprint speed, jumping performance, and lower extremity muscle strength in the study group [23].

In a review study encompassing 21 studies related to sports where postural stability is essential, including team sports such as football, basketball, and hockey, as well as individual sports such as tennis, running, and cycling, Zemkova et al. [35] investigated whether core stability exercises contribute to improving sport-specific performance. They found that the CST

program contributed to significantly improved balanced dynamic visual-motor tasks in football players [35]. The importance of postural control increases due to the intense use of vestibular and proprioceptive information in contact sports. We believe that the increase in ball-kicking velocity detected in the TG after the core stability exercise program is also attributed to dynamic visual-motor development, as demonstrated in the cited study. Barnes et al. [36] stated that runs made during football matches are explosive and cover short distances. They found that 96% and 49% of the runs in football matches were over distances shorter than 30 m and 10 m, respectively. It has been found that the duration of isometric contractions applied in back extension, trunk flexion, and side bridge exercises is related to short-distance sprint and jump performance [36,37]. In a study conducted by Hoskikawa et al. [38], involving 28 male football players aged 12–13, which investigated the effect of a CST program on core muscles and physical performance, the TG followed a core stability exercise program four times a week for six weeks in addition to routine training, whereas the CG followed routine training only. The cross-sectional areas of the core muscles (rectus abdominis/obliques, psoas major, quadratus lumborum, erector spina) were measured using magnetic resonance imaging, and squat and countermovement jump heights were determined. Sprint speed was measured via a 15 m sprint test, and dynamic strength was assessed with an isokinetic dynamometer. Consequently, significant differences were found between the TG and CG in hip extension torque, as well as squat and countermovement jump heights. However, no significant differences were observed in the cross-sectional areas of the core muscles, sprint speed, and hip flexion torque [38].

In comparison, we assessed sprint speed with the 20 m sprint test and identified a statistically significant difference between the TG and CG. We believe that the development of core muscles plays a crucial role in enhancing explosive power. Contrary to our study, Dong et al. [39] reported in their meta-analysis that core exercise training was effective in improving core endurance and balance but not as effective in enhancing sport-specific performance (power, speed, agility, throwing, etc.). Three out of the eight randomized controlled studies included in the meta-analysis were related to football. The period, frequency, and session duration of the core exercise training implemented in these studies ranged from 4 to 12 weeks, 2 to 4 times a week, and 20 to 40 minutes, respectively [39]. We believe that the period and session duration of core exercise training are associated with sport-specific performance parameters. In our study, we initiated the session duration at 40 minutes and extended it to 60 minutes with the transition to the next phase.

In a study evaluating the effect of a specific core exercise program on sprint speed and direction change maneuverability in male football players, Muria and Garrido Gena [15] implemented the core stability exercise program, including football-specific exercises, to the TG, and the standard core stability exercise program to the CG for 20 minutes twice a week for eight weeks. Although they observed more improvement in the TG, they found no significant difference between the groups in sprint speed measured using a 10m sprint test and direction change maneuverability measured using a v-cut test [15]. In a study conducted with elite young football players, Lucano et al. [40] applied a core exercise program to the TG for 20 minutes five times a week for six weeks before the routine training, and a standard warm-up program was applied to the CG. They assessed maximal quadriceps and hamstring muscle performance with an isokinetic dynamometer and jumping performance with a single-leg vertical countermovement jumps test. Consequently, they found a significant increase in the knee extensors' peak torque and knee flexors' peak torque values in the study group, and an improvement in jumping performance in both groups [40]. We believe that the session duration may have played a role in the comparisons between the TG and CG in the aforementioned studies. A similar opinion was expressed by Seaterbakken et al. [41] in their systematic review study. They stated that training aimed at core muscles could increase lower extremity muscle

strength, linear running speed, and direction change/agility performance in young people and adults if applied for more than 18 sessions but less than or equal to 30 minutes. However, to achieve sport-specific performance improvement, this training should be applied for more >30 minutes twice a week. They also stated that there is a need to investigate the effects of different CST programs on sport-specific parameters and physical performance [41].

Doğanay et al. [42] investigated the effects of core exercise training, administered for 35 minutes thrice a week over eight weeks, on running speed, quickness, and agility in U19 male football players. These parameters were measured using the 40 m sprint test, hexagon test, and agility t-test, respectively. Consequently, they observed significant differences between the TG and CG in agility and quickness, but not in sprint speed [42]. In contrast, our study detected significant improvements in ball-kicking velocity and sprint speed, while agility did not show a significant enhancement in adolescent football players. Some studies have speculated that an increase in power, strength, speed, and overall performance may be achieved primarily through participation in football training during the adolescent and preadolescent period [43,44]. In one study in the literature, arguing the opposite perspective, Schilling et al. [45] applied a core strength and endurance training program on 10 college students who did not participate in sports activities. The program was conducted twice a week for six weeks. Significant changes were observed in back extensor endurance, flexor endurance, and lateral musculature endurance values after the completion of the training program. However, no significant improvements were found in sprint speed, agility, and vertical jump performances. As a result, the researchers concluded that standalone strength training does not enhance performance parameters [45]. Similarly, Nesser and Lee [46] asserted that core strengthening does not lead to improvements in sports performance parameters such as agility and running [46].

Agility is defined as the ability to move the body between two points and change direction as easily, quickly, fluently, and controlled as possible, with balance, speed, strength, and nerve-muscle coordination [47]. We measured physical component of agility performance using the 505 agility test. While intra-group evaluations revealed significant changes in agility in both the TG and CG, inter-group comparisons showed no significant differences between the groups. We attribute the increase in agility in both the TG and CG to the fact that the study was carried out during the season, and both groups followed routine training. Additionally, we believe that the reason the study group did not show superiority to the control group in agility performance is related to the absence of agility training specific to football in the CST program.

As a matter of fact, studies conducted in other sports have yielded varied results regarding the impact of core exercises on sport-specific parameters. In one such study examining the effects of core strengthening training on physical and athletic performance in elite handball players, significant improvements were observed in physical components, but not in handball-specific athletic performance [14]. Similarly, a study investigating the effects of core exercises on functional movement patterns in adult tennis players reported significant changes in all functional movement screen test results [48]. Lust et al. [49] also found that a six-week training program, combining open and closed kinetic chain plyometric exercises with core stability exercises, improved core endurance in baseball athletes [49]. Özmen et al. [22] investigated the effect of CST program applied twice a week for six weeks on dynamic balance and agility in adolescent badminton players with a mean age of 10±0.3 years. They measured dynamic balance using the Star Excursion Balance test and agility with the Illinois agility test. Consequently, they observed a significant improvement in dynamic balance in the TG compared to the CG, which followed routine training. On the other hand, they found that agility increased in both groups, but there was no significant difference between the groups [22]. Athletes generally participate in high-intensity training programs along with a core stability program.

Therefore, it is challenging to investigate the effect of a core stability exercise program on performance independently from other training.

Studies conducted in various sports on the effect of CST on sports performance have yielded inconsistent results. The findings of this study indicated that an eight-week core stability exercise program, in addition to routine training, led to improvements in ball-kicking velocity and sprint speed in adolescent male soccer players. However, as no significant difference was observed between the TG and CG in agility performance, we recommend incorporating agility-style exercises into the CST program and extending the program duration for more substantial benefits.

## Limitations

There are several limitations to our study. Firstly, only performance-oriented tests were used for evaluation. The results could have been interpreted more effectively if core strength tests were applied in conjunction with the performance tests. Secondly, adding a new training to the routine program may have increased the load in CST group. Thirdly, the study was conducted during a period when the participants were receiving distance education due to the coronavirus 2019 (COVID-19) measures and were leaving their homes almost exclusively for football training. Control and focus, which are essential for exercise efficiency, are employed during the performance of core exercises. Throughout the study period, it was noted that participants encountered challenges in adaptation attributed to psychosocial factors. Consequently, the application of the study during a period when participants were subjected to pandemic-related restrictions may have influenced the results.

In future research, our intention is to explore the effects of core exercises on injury incidence among adolescent football players and their recovery performance during the rehabilitation period following an injury.

## Practical applications

The study's findings suggest that an 8-week CST program enhanced ball-kicking velocity, sprint speed, and agility performance in adolescent male soccer players. Due to its practicality in the field, affordability of associated equipment costs, and adaptability to routine training programs, we recommend the CST program to football coaches and individuals professionally engaged in football to enhance their performance.

## Supporting information

**S1 Data.**
(XLSX)

## Acknowledgments

The authors gratefully acknowledge the contribution of the participants in this study.

## Author Contributions

**Conceptualization:** Ceyda Sofuoğlu, Volga Bayrakcı Tunay.

**Data curation:** Ceyda Sofuoğlu.

**Formal analysis:** Ceyda Sofuoğlu.

**Investigation:** Ceyda Sofuoğlu, Volga Bayrakcı Tunay.

**Methodology:** Ceyda Sofuoğlu, Volga Bayrakcı Tunay.

**Project administration:** Volga Bayrakcı Tunay.

**Resources:** Ceyda Sofuoğlu, Volga Bayrakcı Tunay.

**Software:** Ceyda Sofuoğlu.

**Supervision:** Zehra Güçhan Topçu, Volga Bayrakcı Tunay.

**Validation:** Volga Bayrakcı Tunay.

**Visualization:** Ceyda Sofuoğlu, Volga Bayrakcı Tunay.

**Writing – original draft:** Ceyda Sofuoğlu.

**Writing – review & editing:** Ceyda Sofuoğlu.

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
