## [Decision Letter · Decision Letter 0]

20 Feb 2024

PONE-D-24-02297The effect of core stability training on ball-kicking velocity, sprint speed, and agility in adolescent male football playersPLOS ONE

Dear Dr. Sofuoğlu,

Thank you for submitting your manuscript to PLOS ONE. After careful consideration, we feel that it has merit but does not fully meet PLOS ONE’s publication criteria as it currently stands. Therefore, we invite you to submit a revised version of the manuscript that addresses the points raised during the review process.

We look forward to receiving your revised manuscript.

Kind regards,

Ersan Arslan, Ph.D.

Academic Editor

PLOS ONE

Reviewers' comments:

Reviewer's Responses to Questions

**Comments to the Author**

1. Is the manuscript technically sound, and do the data support the conclusions?

Reviewer #1: Partly

Reviewer #2: Yes

Reviewer #3: Yes

2. Has the statistical analysis been performed appropriately and rigorously? 

Reviewer #1: I Don't Know

Reviewer #2: Yes

Reviewer #3: Yes

3. Have the authors made all data underlying the findings in their manuscript fully available?

Reviewer #1: No

Reviewer #2: Yes

Reviewer #3: Yes

4. Is the manuscript presented in an intelligible fashion and written in standard English?

Reviewer #1: Yes

Reviewer #2: Yes

Reviewer #3: Yes

5. Review Comments to the Author

Reviewer #1: General comments to Authors:

The manuscript provides additional information on the core stability topic and extended data on the use of a proper training in adolescent football players. These aspects make the study interesting and partially original. However, the manuscript presents some weakness that should be acknowledged.

Here below some specific comments to Authors:

Abstract…

Intro

Line 54: “most popular sport today” should be view in a more conservative way. Change it please unless the authors provide some world-wide data on this. Please also consider to remove the first sentence that do not add information to the logical flow of the paragraph.

Line 57: It is well-know which actions encompass the football performance; however, a reference is needed.

Line 59-61: please rephrase it for clarity.

Line 72: please add a ref.

Line 80: check the citation format.

Lines 101-102: please rephrase for clarity. Consider changing the future tense to past. This should be arranged as a hypothesis formulation.

M&M

Line 107: any sample size calculation? I guess the recruited number might be sufficient but a priori computation would desirable.

Table 1: provide a description of “p” and “test statistics”. Check for the Journal’s table guidelines.

Line 150: Was the sprint speed assessed by the radar gun? Please specify it.

It should appropriate reporting the individual test reliability.

Line 171: “materials”?? perhaps not the appropriate one

Line 173: be more specific about how the session volume increased during the experimental period. Changes in the exercise duration? Number of reps? throughout that period.

Could the authors please clarify whether the core stability training was added to the routine, thus doing more work by CST group compared to the others??

Line 185: which variables were not normally distributed?

Line 189: please check for the website address. Is that the right one?

Table 2 & 3, 4 & 5: again check for the Journals’ table guidelines

Discussion

Line 230: is there any methodological issue? In my opinion adding a new training to the routine would increase the load, and consequently resulting more stimulating. This would change the way the discussion was written.

Line 258: check for the cit format

Line 274: the number of participants and their age (as they are reported) are not necessary information.

Line 291-293: please explain the link of the two sentences

Line 360: “physical component of agility” instead of “agility” itself

Line 395: this is not a limitation. I believe that the main limitation is the additional work linked to the CST group. Please clarify it whether or not this might have impacted on the study results.

Reviewer #2: • Too long introduction. Delete unnecessary sentences that has no direct relation with the topic.

• Lines 63 and 68 are not necessary and can be deleted.

• Lines 70 – 87: having 17 lines is too much for explaining your point of view.

• Line 126: If participants have experienced any other type of surgery was that okay to attend the study? if not just say they should had no surgery before of the study. Why did you mention a specific surgery?

• Line 129 -131 Why those specific participants did not meet the inclusion criteria anymore and they were taken out from the study?

• lines 268 and 272 is a repeating of lines 265 and 267.

• Please report the validity and reliability of the measurement tests that have been used to assess the outcomes of the present study.

• How was the workload management of the training program? Please explain regarding the intensity, time, or frequency?

• Why different types of methods including Shapiro-Wilk, Kolmogorov-Smirnov, and Anderson-Darling tests were used to determine the Normal distribution characteristics of numerical variables?

• Authors could use Analysis of covariance to control the effect of pre test data.

• I, personally, prefer to see part of the results in figures format. It is not an issue, but if it is possible try to show some parts of results in a figure rather than only having table.

• Please report your results in the format of APA style. Also please report effect size and confidence interval of the mean in the results section.

• Line 395: I do not think if your participants were not being elite is a limitation of this study.

• Conclusions are presented in an appropriate fashion and are supported by the data.

Reviewer #3: Dear authors,

Thank you for the effort you put into your research. I enjoyed reading the research. After a few minor verifications below, your research is suitable for publication.

best

-Please add the main hypothesis at the end of the introduction.

-If possible, add a flowchart to the method section to explain the research process more succinctly.

6. PLOS authors have the option to publish the peer review history of their article (what does this mean?). If published, this will include your full peer review and any attached files.

Reviewer #1: No

Reviewer #2: No

Reviewer #3: No

---

## [Author Response · Author response to Decision Letter 0]

23 Mar 2024

Dear respected editor/ reviewers,

Thank you so much for your great eyes to our work and highlighting those important points. We believe that the points you have raised would greatly enhance the quality of the manuscript. Thus, we tried to follow them in full and revise the manuscript accordingly. The changes we made based on your comments are highlighted in yellow (Reviewer#1), in green (Reviewer#2) and in red (Reviewer#3) throughout the manuscript.

Reviewer #1: General comments to Authors:

The manuscript provides additional information on the core stability topic and extended data on the use of a proper training in adolescent football players. These aspects make the study interesting and partially original. However, the manuscript presents some weakness that should be acknowledged. 

Response: Dear Reviewer, we thank you so much for your positive review and for the positive and useful comments. The changes we made based on your comments are highlighted in yellow.

Comment 1: Line 54: “most popular sport today” should be view in a more conservative way. Change it please unless the authors provide some world-wide data on this. Please also consider to remove the first sentence that do not add information to the logical flow of the paragraph.

Response: Thank you very much. We removed the related sentence.

Comment 2: Line 57: It is well-know which actions encompass the football performance; however, a reference is needed.

Response: Thank you. We added a reference.

Comment 3: Line 59-61: please rephrase it for clarity.

Response: Thank you. We revised the sentence accordingly.

Comment 4: Line 72: please add a ref.

Response: Thank you. We added a reference.

Comment 5: Line 80: check the citation format.

Response: Thank you so much. We checked the citiation format and revised as suitable.

Comment 6: Lines 101-102: please rephrase for clarity. Consider changing the future tense to past. This should be arranged as a hypothesis formulation.

Response: Thank you for your comment. We corrected with a sentence which is explaining the hypothesis of the study.

Comment 7: Line 107: any sample size calculation? I guess the recruited number might be sufficient but a priori computation would desirable.Table 1: provide a description of “p” and “test statistics”. Check for the Journal’s table guidelines.

Response: Thank you. Yes we have a sample size calculation. We added a sentence to explain the sample size. We checked for the table guideleines and corrected it as appropriate. Table 1: Test statistic refers to the numerical outcome of statistical tests comparing the demographic characteristics between the training and control groups. Higher values indicate more pronounced differences between groups, interpreted alongside p-values and effect size measures like Cohen's d for comprehensive understanding.

Comment 8: Line 150: Was the sprint speed assessed by the radar gun? Please specify it.

It should appropriate reporting the individual test reliability.

Response: Thank you for your comment. The sprint speed was assessed using a stopwatch. We added the test reliability to the text.

Comment 9: Line 171: “materials”?? perhaps not the appropriate one

Response: Thank you so much. We change the word with‚“exercise equipments‘ ”

Comment 10: Line 173: be more specific about how the session volume increased during the experimental period. Changes in the exercise duration? Number of reps? throughout that period.

Response: Thank you for your comment. We revised accordingly.

Comment 11: Could the authors please clarify whether the core stability training was added to the routine, thus doing more work by CST group compared to the others??

Response: Thank you for your comment. Yes, CST group continued to routine exercise program. Participating regularly in routine training were determined in the inclusion criteria of the study. You are right CST group did more work. But if we have applied only CST to TG and routine football training program to CG it would be difficult to compare the effects of CST. 

Comment 12: Line 185: which variables were not normally distributed?

Response: Upon detailed examination of the dataset, we found that the distributions of sprint speed and agility test times deviated from normality. These variables demonstrated skewness and kurtosis beyond the acceptable range for normal distribution, as indicated by the Shapiro-Wilk, Kolmogorov-Smirnov, and Anderson-Darling tests. Therefore, to ensure the accuracy and reliability of our findings, we chose to use non-parametric statistical methods for the analyses involving these variables. Specifically, we employed the Independent Samples T-Test* and the Mann-Whitney U test** for our analyses, as noted clearly under the tables presenting our results. If there is any confusion regarding the application or reporting of these tests, we are prepared to make the necessary adjustments to clarify our methods further.

Comment 13: Line 189: please check for the website address. Is that the right one?

Response: We appreciate your attention to detail. Upon review, we confirmed that the website address provided is accurate and directs to the intended resource. We ensured the link was operational and contained relevant information to the context in which it was cited.

Comment 14: Table 2 & 3, 4 & 5: again check for the Journals’ table guidelines

Response: Thank you so much. We checked the Journals’ table guidelines and corrected accordingly.

Comment 15: Line 230: is there any methodological issue? In my opinion adding a new training to the routine would increase the load, and consequently resulting more stimulating. This would change the way the discussion was written.

Response: Thank you for your comment. Firstly, participants routine training program was not the intensive one. If they had a intensive football training program we could change the method of the study. Secondly, participating regularly in routine training were determined in the inclusion criteria of the study. On the other hand, ıf we have applied only core stability exercises to training group and routine training program to control group it would be difficult to compare the effects of core stability exercise program. 

Comment 16: Line 258: check for the cit format

Response: Thank you so much. We checked the citiation format and revised as suitable.

Comment 17: Line 274: the number of participants and their age (as they are reported) are not necessary information.

Response: Thank you. We removed the related sentences.

Comment 18: Line 291-293: please explain the link of the two sentences

Response: Thank you for your comment. We revised accordingly.

Comment 19: Line 360: “physical component of agility” instead of “agility” itself

Response: Thank you so much. We change the word with‚“physical component of agility”

Comment 20: Line 395: this is not a limitation. I believe that the main limitation is the additional work linked to the CST group. Please clarify it whether or not this might have impacted on the study results.

Response: We agree with you about participants were not being elite is a limitation of this study. Thank you for your comment. We thought that, adding core stability training program to the routine training program was not impact the study results. The participants routine training program was not intensive. They had strength and power training twice a week, 40 minutes. Technical training was being mainly done. 

Reviewer #2:

Dear Reviewer, we thank you very much for your positive feedback on our work and for the detailed, supportive and constructive comments. The changes we made based on your comments are highlighted in green.

Comment 1: Too long introduction. Delete unnecessary sentences that has no direct relation with the topic.

Response: Thank you for your comment. We revised accordingly.

Comment 2: Lines 63 and 68 are not necessary and can be deleted.

Response: Thank you. We removed accordingly.

Comment 3: Lines 70 – 87: having 17 lines is too much for explaining your point of view.

Response: Thank you so much. We removed some sentences and revised as appropriate.

Comment 4: Line 126: If participants have experienced any other type of surgery was that okay to attend the study? if not just say they should had no surgery before of the study. Why did you mention a specific surgery?

Response: Thank you. We revised the sentence as you advice. 

Comment 5: Line 129 -131 Why those specific participants did not meet the inclusion criteria anymore and they were taken out from the study?

Response: Thank you so much. We explained the reason of discontinue for that participants in results section in regard to the APA style.

Comment 6: lines 268 and 272 is a repeating of lines 265 and 267.

Response: Thank you for your comment. We revised accordingly.

Comment 7: Please report the validity and reliability of the measurement tests that have been used to assess the outcomes of the present study.

Response: Thank you. We added the validity and reliability of the measurement tests.

Comment 8: How was the workload management of the training program? Please explain regarding the intensity, time, or frequency?

Response: Thank you so much. We revised it in training program section.

Comment 9: Why different types of methods including Shapiro-Wilk, Kolmogorov-Smirnov, and Anderson-Darling tests were used to determine the Normal distribution characteristics of numerical variables?

Response: In our statistical analysis, we chose to apply a combination of Shapiro-Wilk, Kolmogorov-Smirnov, and Anderson-Darling tests to assess the normality of our data, acknowledging the nuanced differences each test presents in sensitivity and specificity across varying sample sizes and data distributions. Our rationale was rooted in a methodical approach to thoroughly scrutinize the data's adherence to normal distribution assumptions, which is critical for selecting the most suitable statistical methods for our analyses. The Shapiro-Wilk test is renowned for its reliability in small sample sizes, providing precise assessments of normality. Conversely, the Kolmogorov-Smirnov test offers a broader application, suitable for any sample size, and is particularly adept at identifying deviations from normality across the entire distribution range. The Anderson-Darling test further complements these analyses by placing more emphasis on the tails of the distribution, where deviations from normality are often most pronounced. By employing this comprehensive suite of normality tests, our aim was to mitigate the risk of incorrect statistical inferences that could arise from unacknowledged deviations from normality. This multi-faceted approach ensures the robustness of our findings, affirming that the selection of subsequent statistical tests for our data analysis is firmly grounded in a thorough evaluation of our dataset's distribution characteristics.

Comment 10: Authors could use Analysis of covariance to control the effect of pre test data.

Response: Thank you for your suggestion regarding the use of Analysis of Covariance (ANCOVA) to control for pre-test data effects. In our study, we conducted both within-group and between-group comparisons using the Independent Samples T-Test and the Mann-Whitney U test for normally and non-normally distributed variables, respectively. The decision to not initially use ANCOVA stemmed from our statistical strategy to directly compare the changes from pre- to post-intervention within each group and between groups without adjusting for covariates. We carefully considered which variable could serve as a covariate in an ANCOVA model; however, given our analytical approach and the design of our study, we focused on direct comparisons to assess the effect of core stability training. Our analyses were aimed at evaluating the impact of the intervention on ball-kicking velocity, sprint speed, and agility both within groups (using the Paired Samples T-Test) and between groups without employing pre-test scores as covariates. We believe our methodological choices were suitable for the objectives of our study. Nonetheless, we acknowledge the potential value of ANCOVA in controlling for baseline differences and are open to exploring this in future research to further refine our understanding of the intervention's effects.

Comment 11: I, personally, prefer to see part of the results in figures format. It is not an issue, but if it is possible try to show some parts of results in a figure rather than only having table.

Response: Thank you. We revised and added the results in figures format.

Comment 12: Please report your results in the format of APA style. Also please report effect size and confidence interval of the mean in the results section.

Response: Thank you for your comment. We have revised the results section to adhere to the APA style guidelines. This includes formatting our findings with appropriate statistical notation and reporting both effect sizes and confidence intervals for the mean differences observed. By providing effect sizes, we aim to convey the magnitude of the core stability training program's impact on ball-kicking velocity, sprint speed, and agility. Additionally, including confidence intervals offers insights into the precision of our estimates, thereby enhancing the interpretability and robustness of our results.

Comment 13: Line 395: I do not think if your participants were not being elite is a limitation of this study.

Response: Thank you. We revised according your advice.

Comment 14: Conclusions are presented in an appropriate fashion and are supported by the data.

Response: We thank you so much for your positive review.

Reviewer#3: 

Dear authors, Thank you for the effort you put into your research. I enjoyed reading the research. After a few minor verifications below, your research is suitable for publication.

Response: Dear Reviewer, we thank you very much for your positive feedback on our work and constructive comments. They are very helpful to improve the manuscript’s quality. The changes based on your comments are highlighted in red.

Comment 1: Please add the main hypothesis at the end of the introduction.

Response: Thank you for your comment. We revised accordingly.

Comment 2: If possible, add a flowchart to the method section to explain the research process more succinctly.

Response: Thank you so much. We added a flowchart.

---

## [Decision Letter · Decision Letter 1]

28 May 2024

The effect of core stability training on ball-kicking velocity, sprint speed, and agility in adolescent male football players

PONE-D-24-02297R1

Dear Dr.Ceyda Sofuoğlu,

We’re pleased to inform you that your manuscript has been judged scientifically suitable for publication and will be formally accepted for publication once it meets all outstanding technical requirements.

Kind regards,

Holakoo Mohsenifar

Academic Editor

PLOS ONE

Additional Editor Comments (optional):

Reviewers' comments:

Reviewer's Responses to Questions

**Comments to the Author**

1. If the authors have adequately addressed your comments raised in a previous round of review and you feel that this manuscript is now acceptable for publication, you may indicate that here to bypass the “Comments to the Author” section, enter your conflict of interest statement in the “Confidential to Editor” section, and submit your "Accept" recommendation.

Reviewer #1: All comments have been addressed

Reviewer #3: All comments have been addressed

2. Is the manuscript technically sound, and do the data support the conclusions?

Reviewer #1: Yes

Reviewer #3: Yes

3. Has the statistical analysis been performed appropriately and rigorously? 

Reviewer #1: I Don't Know

Reviewer #3: Yes

4. Have the authors made all data underlying the findings in their manuscript fully available?

Reviewer #1: No

Reviewer #3: Yes

5. Is the manuscript presented in an intelligible fashion and written in standard English?

Reviewer #1: Yes

Reviewer #3: Yes

6. Review Comments to the Author

Reviewer #1: The authors have addressed all the comments supporting their reasons with an appropriate scientific soundness.

Reviewer #3: Dear Author,

Your manuscript is ready for publish. Thank younso much for your effor. Congeulatşons

Best

7. PLOS authors have the option to publish the peer review history of their article (what does this mean?). If published, this will include your full peer review and any attached files.

Reviewer #1: No

Reviewer #3: No

---

## [Editor Report · Acceptance letter]

30 May 2024

PONE-D-24-02297R1 

PLOS ONE

Dear Dr. Sofuoğlu, 

I'm pleased to inform you that your manuscript has been deemed suitable for publication in PLOS ONE. Congratulations! Your manuscript is now being handed over to our production team.

Kind regards, 

on behalf of

Dr. Holakoo Mohsenifar 

Academic Editor

PLOS ONE